# Recent Advances in the Treatment and Supportive Care of POEMS Syndrome

**DOI:** 10.3390/jcm11237011

**Published:** 2022-11-27

**Authors:** Maroun Bou Zerdan, Tracy I. George, Silvia Tse Bunting, Chakra P. Chaulagain

**Affiliations:** 1Department of Hematology-Oncology, Myeloma and Amyloidosis Program, Maroone Cancer Center, Cleveland Clinic Florida 2950, Weston, FL 33331, USA; 2Department of Internal Medicine, SUNY Upstate Medical University Hospital, Syracuse, NY 13210, USA; 3Department of Pathology, University of Utah School of Medicine, Salt Lake City, UT 84132, USA; 4Division of Hematopathology and Flow Cytometry Laboratory, Department of Pathology, Cleveland Clinic Florida 2950, Weston, FL 33331, USA

**Keywords:** supportive care, POEMS, pathogenesis, treatment

## Abstract

POEMS is a rare clonal plasma cell disorder characterized by multi-systemic features that include demyelinating peripheral neuropathy, organomegaly, endocrinopathy, presence of monoclonal proteins (M-protein), and skin changes. Even though the pathophysiology is poorly understood, recent studies suggest that both clonal and polyclonal plasmacytosis leading to the production of pro-inflammatory cytokines and angiogenic mediators play the central role. These mediators including vascular endothelial growth factor (VEGF) are the driving forces of the syndrome. The diagnosis of POEMS is not always straight forward and often the diagnosis is delayed. It is based on fulfilling mandatory criteria of polyradiculoneuropathy and monoclonal protein and the presence of one major criterion (Castleman disease, sclerotic bone lesions, or elevated VEGF), and at least one minor criterion. Due to the presence of neuropathy, it can be confused with chronic inflammatory demyelinating polyradiculopathy (CIDP), and if thrombocytosis and splenomegaly are present, it can be confused with myeloproliferative neoplasms. Due to the rarity of the syndrome, clear guidelines for treatment are still lacking. Immediate treatment targeting the underlying plasma cell proliferation results in a dramatic response in most patients. The key is early diagnosis and immediate anti-plasma cell directed therapy for the best clinical outcomes. For patients with disseminated disease as defined by bone marrow involvement or more than three osteosclerotic bone lesions, high-dose chemotherapy with autologous hematopoietic stem cell transplant (ASCT) yields durable responses and is the preferred treatment in eligible patients. For patients with localized bony disease, radiotherapy has proven to be very effective. Lenalidomide and dexamethasone is a proven therapy in patients ineligible for ASCT. In this review article, we tackle the diagnostic approach and discuss the latest treatment modalities of this rare debilitating disease.

## 1. Introduction

Osteosclerotic myeloma (OM) [1], Takatsuki syndrome [2], or Crow-Fukase syndrome [3] are rare entity characterized by a single or multiple bony lesions, often associated with a rare paraneoplastic syndrome: POEMS. An estimated, one third of patients with OM have neuropathy and half the patients with myeloma and neuropathy have osteosclerotic lesions [4]. Clinicians consider, as of recently, that OM and POEMS are along one spectrum of plasma cells dyscrasias [5]. One of the first associations proposed between OM and POEMS was by Crows et al. back in the 1950s when he published an article describing two case reports presenting with osteosclerotic plasmacytoma associated with neuritis, odd skin changes, mild lymphadenopathy, and ankle weakness [3].

The name of this syndrome is a mnemonic of the clinical features of patients: P for polyneuropathy, O for Organomegaly, E for Endocrinopathy, M for Monoclonal (M)-proteins, and S for Skin changes. As the mnemonic suggests, it is a multisystem disorder [1,6,7]. When diagnosed and treated early, the median survival of patients is 13.7 years [4]. Yet, if the disease is left untreated, it can progress to affect the respiratory and nervous systems at a higher degree and can be fatal.

## 2. Pathogenesis of the Disease

The pathogenesis of POEMS syndrome is complex and not fully understood. It has been suggested that it is caused by two entities: the expansion of clonal and polyclonal plasma cells and the surge of pro-inflammatory and angiogenic cytokines. The messenger ribonucleic acid (mRNA) levels of vascular endothelial growth factor (VEGF) are higher in CD138 positive plasma cells derived from the bone marrow relative to the CD138 negative plasma cells. Interestingly, when Wang et al. immunophenotyped the cells considered to be the major source of serum VEGF, these cells were polyclonal in nature confirming that the polyclonal plasma cells in POEMS play an important role for production of these cytokines leading to the constellation of signs and symptoms [8]. However, when Isshiki et al. utilized single-cell RNA-Seq of bone marrow plasma cells from patients with POEMS syndrome, VEGF mRNA was not upregulated in POEMS clones, directly indicating that serum VEGF is not produced by POEMS clones [8]. In comparison, monoclonal plasma cells in POEMS, exhibited an increase in the production intracellular interleukin-6 (IL-6) expression, a pro-inflammatory cytokine and inducer of VEGF expression and secretion from the cells [7,9]. Hence, pathogenesis is an interplay between monoclonal and polyclonal plasma cells in POEMS.

Recently, treatments that target the plasma cells producing the M-protein have shown excellent clinical responses. This suggests that the M-protein might be one of the driving factors of the disease. The plasma cell clone producing the M protein is usually λ light chain restricted, but atypically can also present with Kappa light chain restriction [10,11]. Clonal immunoglobulin λ light variable chain (IGLV) was strictly derived from *IVLG-1–40* and *IVLG-1–44* genes. Additionally, patients with an M-protein restricted to *IVLG-1–40* experienced severe clinical symptoms [12]. In some cases, IGLV gene rearrangement clone was not significantly increased suggesting a small number of monoclonal plasma cells responsible for producing this large serum concentration of monoclonal proteins. On the other hand, the size of the dominant clone is decreased and increased with disease remission and relapse, respectively [13].

The serum VEGF level is elevated in most patients with POEMS. This multifunctional cytokine is produced physiologically by cells derived from the bone marrow such as plasma cells, osteoblasts, and platelets [14,15]. Its role is pivotal in the regulation of angiogenesis, microvascular permeability, and migration of hematopoietic stem cells. Its main target is the VEGFR1 receptors on endothelial cells. The exaggeration of these physiologic effects might be related to the full clinical presentation of POEMS including volume overload, papilledema, and hemangiomata. Its high levels are suspected to be the cause of an increase in the intraneural pressure leading to edema and vascular permeability, which may result in nerve damage [16], instead of direct neuronal invasion [17]. However, patients treated with bevacizumab, an anti-VEGF monoclonal antibody, did not show clinical improvement. VEGF is induced upstream by the hypoxia transcription factor-1 (HTP-1) [16]. Specifically, the nerve VEGF was highly expressed in blood vessels and non-myelin forming Schwann cells in patients with POEMS syndrome.

The probability to have POEMS gets higher with greater serum VEFG levels [18,19]. Serum VEFG decreases rapidly post-treatment and usually takes six months to stabilize back to baseline. Add to that, patients with normalized serum VEGF were relapse free and had a better overall clinical improvement. The rate at which serum VEGF was reduced in the first six months of therapy correlated with an increase in grip strength, and a compound muscle action amplitude at 12 months post treatment [20]. Note that other cytokines such as tumor necrosis factor alpha (TNF-α), Il-6, and Il-12 are also involved in disease progression [16,17,21,22] but the levels of these cytokines are not routinely measured and followed up in clinical practice.

## 3. Clinical Presentation and Diagnosis

The diagnosis of POEMS is mostly clinical, complemented by laboratory studies and radiographic imaging. Any sign or symptom should elicit an in-depth investigation for POEMS particularly in a patient with monoclonal gammopathy and peripheral neuropathy. To diagnose POEMS mandatory criteria are required, along with one major and one minor criteria [23]. The cardinal criteria are polyradiculoneuropathy [24] and plasma cell proliferative disorder causing monoclonal gammopathy (Table 1).

On laboratory findings, Wang et al., identified a new marker for POEMS syndrome: N-terminal of type I collagen with a cut-off of 70 ng/mL [31] but it has not been used in clinical practice yet. Patients with POEMS are usually at an increased risk of venous or arterial thrombosis (likely due to the thrombogenic potential of elevated VEGF and due to an increase in fibrinopeptide A and thrombin-antithrombin complexes), with nearly 20–30% presenting with one of these complications [32,33]. The serum VEGF is markedly elevated and correlates with the activity of the disease [7,14,19]. Other cytokines such as IL-1β, TNF-α, and IL-6 are often increased as well, but the high level of serum VEGF is highly suggestive of POEMS. Measuring VEGF levels is an essential component of POEMS initial work up and to follow up on response to therapy and disease activity.

Classic bone marrow findings have been described and are shown in Figure 1.

Oftentimes the bone marrow plasma cell clone is of small size (~5% in 50% of patients), typically lambda monotypic with concomitant polyclonal plasmacytosis and the clonal plasma cell proliferation can be detected by bone marrow biopsy in ~66% of patients. Reactive non-clonal lymphoid aggregates are frequently present, and clustering and hyperplasia of megakaryocytes are observed in POEMS patients with peripheral thrombocytosis. In one study, megakaryocytes clustering with high expression of VEGF as well as lymphoid aggregates was present in 94% of the cases, which was not the case in patients with MGUS, MM, or amyloidosis [34]. Typically, in POEMS, sensory neuropathy occurs first followed by motor neuropathy over time causing distal weakness and even atrophy (foot drop). Dysautonomia, which is a feature of systemic AL amyloidosis, is uncommon with POEMS. Nerve conduction studies show slowing of nerve conduction which suggests demyelination in the nerve trunk rather than the terminals. Though a routine nerve biopsy is not needed in clinical practice but on nerve biopsy, demyelination is often axonal, favoring lower limbs which will help in the differentiation of POEMS from CIDP [35,36,37,38,39]. Diagnostic work up should also include computed tomography (CT) or magnetic resonance imaging (MRI) to detect bone lesions as the whole-body bone survey radiography may miss the smaller focal osteoclerotic bone lesions. Positron emission tomography (PET) with CT scan may demonstrate fluorodeoxyglucose (FDG)-avid bone lesions which can be targeted for core needle biopsy for diagnostic purposes. Serum and urine electrophoresis with immunofixation and serum free light chain levels need to be measured. Screening for endocrinopathies should be achieved by measuring appropriate hormone levels, and funduscopic examination should be completed in all patients regardless of the presence or absence of ocular symptoms.

The differential diagnoses to keep in mind are chronic inflammatory polyradiculoneuropathy (CIPD), anti-myelin-associated-glycoprotein (MAG) neuropathy, monoclonal gammopathy of undetermined significance (MGUS), and immunoglobulin light chain (AL) amyloid neuropathy. Any differential diagnosis where the patient does not respond to the usual treatment (e.g., for CIDP) should elicit further investigation for POEMS syndrome. Due to the debilitating nature of POEMS syndrome, early detection and intervention is critical to mitigate the risk of permanent neurologic disability.

Last, it is also important to note that He et al. described a group of patients with clinical manifestations highly resembling POEMS syndrome except for no evidence of monoclonal gammopathy [40]. In these patients, elevated VEGF levels may help in teasing out such variants of POEMS syndrome [40]. Thus, a variant form of POEMS syndrome arises. This was suggested according to characteristic constellations, elevated VEGF levels, and responsiveness to therapies directly targeting plasma cells [40].

## 4. Treatment

Due to the rarity of this disease, there is a lack of clear evidence-based guidelines for treatment. Clinicians use therapeutic strategies mostly based on retrospective case series of POEMS syndrome with proven efficacy in MM and systemic AL amyloidosis (Figure 2).

Back in the 1990s, the treatment of POEMS syndrome shifted towards suppressing plasma cell proliferation and secretion of pro-inflammatory and pro-angiogenic cytokines.

The treatment of POEMS syndrome can be divided into two major categories: targeting the clone by specific anti-plasma cell directed therapy and targeting the symptoms with symptomatic and supportive care. To have a better overall outcome, physicians aim to incorporate both strategies in their treatment plan. During treatment, hematologic response is monitored by measuring the M-protein concentration and the serum VEGF levels. Clinical response is most of the times dissociated from the hematological response [41,42]. Serum immunoglobulin free light chains are elevated, with the exception of 18% of the patients, making this test also of limited value in the follow up [43]. The most efficacious strategy is taking the baseline parameters assessing them every visit or every other visit [44].

## 5. Targeting the Clone

The role of radiotherapy

This strategy relies on the presence or absence of bone marrow involvement upon bone marrow aspiration and biopsy and the number of bone lesions seen in the imaging studies [45,46]. If diagnostic imaging tests (CT scan, MRI, or PET scan) reveal 1–3 bone lesions with no bone marrow involvement then radiation therapy is the first line of therapy with excellent long-term outcomes [47,48,49]. The main goal of radiation therapy is to ablate the boney lesions without significant toxicity. A suggested dose of radiotherapy between 30 and 54 Gy (median dose ~45Gy) can achieve a high cure rate of approximately 82% [50]. If the patient is rapidly deteriorating, simultaneous use of corticosteroids, e.g., dexamethasone (40 mg/day 1–4 times a week for 2 weeks) or prednisone (1 mg/kg daily for two weeks) is a reasonable adjuvant therapy to radiotherapy. Corticosteroids can be tapered gradually over the next months with follow up and vigilance for adrenal insufficiency [45,46]. In most patients, clinical improvement is seen in approximately 3 months post-treatment but may take up to 36 months post-therapy with neuropathic symptoms taking the longest for improvement. Patients with more than three skeletal lesions are best treated with systemic anti-plasma cell therapy. Similarly, approximately 66% of patients with POEMS syndrome have bone marrow involvement at diagnosis and are treated with systemic anti-plasma cell therapy (Figure 2). In this scenario, radiation therapy alone is not effective, and it should be combined with systemic anti-plasma cell chemotherapy or autologous stem cell transplant (ASCT) after high dose melphalan chemotherapy.

b.Autologous Stem Cell Transplant (ASCT)

In cases of bone marrow involvement or more than three bone lesions, melphalan based conditioning followed by ASCT is considered the first line treatment by many; given the patient is able to tolerate the treatment based on functional status and adequate organs function [42]. Several retrospective studies and case series support the role of upfront ASCT in eligible patients [51]. It has been proven effective in improving neuropathy decrease in morbidity, disability, and in improving quality of life [52]. For patients undergoing ASCT, high dose melphalan (140–200 mg/m^2^) is used as conditioning regimen. 

Induction therapy may reduce the incidence of complication [53]. Normalization of serum VEGF level at ASCT may prolong OS and PFS [54,55,56,57,58,59,60,61,62,63,64,65,66,67].

In contrast with the ASCT in MM and AL amyloidosis, the patients with POEMS syndrome are at a high risk of engraftment syndrome (ES), in approximately 50% of the cases, and respond well to corticosteroid steroid therapy. ES is characterized by a fever (most common symptom), rash, diarrhea, weight gain and respiratory symptoms which can be severe enough to cause respiratory failure needing mechanical ventilation. This can occur between day 7 and 15 post stem cell infusion with a higher mortality rate compared to patients with MM undergoing the same treatment (7.4% vs. 2%, respectively) [42]. A positive correlation was found between splenomegaly/lymphadenopathy and risk of ES. It is worth mentioning that POEMS patients typically have a delayed neutrophil engrafting (median of 16 days) and may require platelets and erythrocytes transfusions [41]. The use of granulocyte colony stimulating factor (G-CSF) solely had major adverse events and was associated with an increase in vascular permeability and fluid overload [17]. Pre-ASCT treatment reduces the incidence of ES among patients who already had decreased level of VEGF [68].

Overall, ASCT is a safe and effective choice in many patients and can provide durable response in about 90% of eligible patients. Different studies and retrospective analysis confirmed high rates of overall survival (OS) and progression free survival (PFS), 98% at one year, following ASCT [51,52,69,70], with a significantly lower rate of relapse [69,70] in a median of 5 months post treatment [68,71]. Relapses of patients after ASCT is positively correlated with immunoglobulin G-λ monoclonal gammopathy presence of FDG-avid lesions on baseline PET scan and inability to achieve hematologic response with ASCT [72]. Persistent or relapse disease post-ASCT can be treated with anti-plasma cell therapy such as lenalidomide.

c.Immunomodulatory Drugs (IMiDs): Lenalidomide and thalidomide therapy

Given the anti-angiogenic effect of IMiD, such as thalidomide or lenalidomide, and their effectiveness in the treatment of MM, they were an option to consider in POEMS syndrome. Thalidomide and bortezomib should be used with caution due to their potential to induced neuropathy or worsen pre-existing neuropathy [73]. For that reason, lenalidomide is the IMiD of choice in POEMS syndrome where neuropathy is the feature of the disease itself [30,55,74,75]. Lenalidomide has been shown to be more potent than thalidomide in its direct apoptotic effect against myeloma cells, as well as in inhibition of inflammatory cytokine production and stimulation of T cells and NK cells [76]. It does not cause sedation, bradycardia, or constipation, and the incidence of neuropathy is lower [77]. Both thalidomide and lenalidomide dare associated with increased incidence of deep-vein thrombosis (DVT) and thromboembolism, especially when combined with anthracycline and dexamethasone [78]. In a study by Gay et al., patients receiving lenalidomide/dexamethasone had longer time to progression (median, 27.4 vs. 17.2 months; *p* = 0.019), PFS (median, 26.7 vs. 17.1 months; *p* = 0.036) and OS (median not reached vs. 57.2 months; *p* = 0.018) than patients receiving lenalidomide/dexamethasone [78]. Lenalidomide/dexamethasone appears well-tolerated and more effective than thalidomide/dexamethasone [78]. Randomized trials are needed to confirm these results.

Patients treated with lenalidomide experienced systemic improvement, including a decrease in the severity of polyneuropathy, fluid overload and organomegaly. A decrease in the monoclonal protein and VEGF levels was observed as well [79,80,81]. The neuropathy relief was the most important advance in terms of the quality of life of the patients [41,45,82,83]. Patients who presented at baseline with hypothyroidism had an inferior OS than euthyroid patients even though the combination of lenalidomide and corticosteroids improved their overall thyroid function [81,84,85,86,87,88].

The recommended treatment is lenalidomide (15–25 mg orally) taken days 1–21 every 28 days with weekly dexamethasone 20–40 mg. These patients had a dramatic improvement up to one-year post-treatment [83]. An open-label prospective study using lenalidomide with dexamethasone regimen (LDex) in patients who were also ineligible to undergo ASCT was conducted. More than two-thirds (72%) responded to treatment and had clinical and neurological improvement with low toxicity rates. At 39 months of follow up post therapy, all patients were alive and PFS was 59% at 3 years of follow up [86]. Approximately 30% of POEMS patients have arterial and venous thrombosis, a number significantly higher than originally reported [89]. A patient at risk of thrombosis, bleeding or of falling must be assessed when taking the decision to initiate LDex. All IMiDs including lenalidomide have thrombogenic potential. Unless there is a contraindication thromboprophylaxis is recommended at least with aspirin therapy or even full anticoagulation depending on the thrombotic risks associated with a particular patient. A randomized, double-blinded, placebo-controlled, phase 2/3 trial in Japan of patients not eligible for ASCT was performed using either oral thalidomide or placebo. Serum VEGF was significantly reduced by thalidomide therapy with a high risk of cardiotoxicity and bradycardia. 

A Phase II multicenter trial (NCT02921893) in Mayo clinic, Rochester, United states is recruiting patients to test different combination of treatments randomizing patients into two groups. One arm will receive ixazomib citrate (a drug that is approved for MM and works by proteasome inhibition), with lenalidomide, dexamethasone, and ASCT, and another arm will receive ixazomib citrate, lenalidomide, and dexamethasone only. Patients will be observed for toxicity and adverse events as well as the difference between the two arms in the efficacy of normalizing serum VEGF and hematologic response.

d.Melphalan and corticosteroid

Melphalan combined with dexamethasone (MDex) is also used in different plasma cell disorders [90]. The first prospective study tackling the treatment of patients newly diagnosed with POEMS syndrome showed an efficacy and low toxicity with MDex combination, 10 mg/m^2^ of melphalan added to dexamethasone 40 mg/day on days 1 to 4 of every 28-day cycle, then ASCT vs. another arm of patients who received only MDex. An 80.6% of patients had a hematologic response and 77.4% had a neurologic response at 3 months post initiation of therapy with a median to maximal response of 12 months. Patients treated with MDex had a substantial improvement in the serum VEGF, relief from organomegaly, extravascular fluid overload, and pulmonary hypertension, with a median PFS of 21 months. Interestingly, despite the hematologic response, both arms had a neurologic response to MDex therapy. In comparison, another study performed in Japan by Suichi et al. in 2019, showed a 100% improvement rate of neurological signs after 1 to 3 months post ASCT [91]. MDex was 100% effective in neurological signs with a relatively short initial response of 3 months compared to the ASCT with less toxicity and side effects.

Younger patients need a longer follow up period to assess long term complications of oral melphalan therapy because it is associated with myelodysplasia (MDS) as well as therapy related acute myeloid leukemia (AML) [42]. Furthermore, damage of stem cells is a possibility so it might not be always suitable to treat patients with it prior to ASCT [92,93]. MDex is not used in routine clinical practice in the USA due to the availability and effectiveness of other therapies such as lenalidomide based or bortezomib-based therapies.

e.Bortezomib and dexamethasone

Bortezomib, a proteasome inhibitor, inhibits the secretion of VEGF and IL-6, stops angiogenesis, and induces B cell apoptosis in a dose-dependent manner [93]. The combination of cyclophosphamide, bortezomib, and dexamethasone (CyborD or VCD) had a high efficacy in treatment but due to presence of baseline neuropathy in most patients with POEMS syndrome, bortezomib based therapy may not be the first preferred therapy [94,95]. Due to this reason, we prefer lenalidomide plus dexamethasone over bortezomib based therapy in POEMS syndrome. The major adverse events with bortezomib based therapy in POEMS syndrome was painful neuropathy which can be reduced (but not eliminated) with the subcutaneous administration of the medication. Yet, the risk is there, and it is especially increased to 50% after a cumulative dose ≥45 mg/m^2^ [96]. Bortezomib can help in decreasing pleural effusions, ascites, and pulmonary hypertension in POEMS. Case reports highlight the use of BorDex therapy with radiation or ASCT that resulted in a dramatic improvement in serum VEGF as well as peripheral neuropathy without significant toxicity [97]. In patients relapsed or resistant to treatment including ASCT, lenalidomide or bortezomib regimens as a salvage treatment showed significant improvement [98].

In one report, bortezomib-dexamethasone improved peripheral neuropathy and resulted in a complete response. The regimen was well tolerated in many patients with only a few reported adverse effects, and it was adequate for collection of hematopoietic stem cells for ASCT [99] when used for induction therapy pre-ASCT. 

f.Monoclonal Antibody Therapy
Anti-CD38 monoclonal antibodies (Daratumumab and Isatuximab)Daratumumab combined with lenalidomide in relapsed MM was proven effective [100,101] and the responses to daratumumab in light chain amyloidosis have been well documented [102]. In patients with POEMS syndrome, the response to daratumumab and lenalidomide was remarkable as well based on case reports. This suggests that this combination is a promising less toxic alternative to ASCT that warrants further exploration in both newly diagnosed and relapsed/refractory patients [103]. Importantly, this regimen provides rapid, deep and durable responses and restores their functionality [104]. This regimen was suggested in multidrug resistant patients but needs further evaluation using prospective studies [103]. A phase II clinical trial (NCT04396496) at the University of Arkansas in the United States is currently investigating the effects of daratumumab combined with lenalidomide (up to 12 four-week cycles) with the primary objective being the outcome of neuropathy and performance status of patients. Another anti-CD38 immunotherapy isatuximab which is approved for relapsed/refractory MM may also be effective in the treatment of POEMS syndrome but its use in POEMS syndrome has not yet been reported. Anti-VEGF monoclonal antibody (Bevacizumab)In theory the use of bevacizumab was appealing in many cases [19,74,104,105,106,107,108]. The benefit was noticed in patients who received bevacizumab during alkylator therapy or preceding therapy [105,106]. The treatment was combined with ASCT or high dose chemotherapy [70,74] or with radiation therapy and cyclophosphamide. Clinical response and anti-VEGF response within 6 months of therapy were observed [86]. Yet, other reports where the patients were treated with bevacizumab died shortly after treatment [68,107,108]. When bevacizumab was used solely, serum VEGF decreased rapidly. However, the number of patients included in different studies was small, and a significant proportion of patients did not respond or died during treatment [108,109,110]. Due to lack of efficacy and potential for increasing mortality in some reports, bevacizumab should not be used in clinical practice for treatment of POEMS syndrome. Chimeric antigen receptor T (CAR-T) therapyBecause the treatments for MM were expected to be useful in many patients with POEMS, chimeric antigen receptor T (CAR-T) cells targeting B cell maturation antigen (BCMA) have been used in the treatment of relapsed and refractory multiple myeloma (RRMM) with excellent response. Recently, a report of treatment by anti-BCMA CAR-T cells in POEMS demonstrated that the treatment may be a feasible therapeutic option for patients with POEMS syndrome and RRMM who do not respond well to traditional therapies [111]. We caution against routine referral of POEMS syndrome for CAR-T therapy because the experience so far is very limited.Comparison of all regimensA retrospective analysis of 347 patients compared ASCT to MDex and LDex groups. ASCT was well tolerated and had the highest response and PFS rates out of all three regimens [70]. One explanation is that ASCT eradicates the underlying plasma cell clones. Yet, patients undergoing ASCT were younger with better organ function and therefore lower risk patients compared to MDex and LDex groups who were typically higher risks not eligible for ASCT. No significant difference in OS was observed between all three treatments. MDex therapy in low-risk patients achieved a response similar to that of ASCT, but that was not the case in patients categorized as medium or high-risk. LDex was highly effective in patients newly diagnosed with a similar response rate and OS as those treated with ASCT. PFS was inferior in high-risk patients even after ASCT therapy. Therefore, utilizing LDex for newly diagnosed patients is a consideration especially those who do not want ASCT or are not candidate for it [112].


## 6. Targeting the Symptoms of the Disease

Peripheral neuropathy

Neuropathy of POEMS as previously described is peripheral, symmetrical, ascending in nature. It involves the sensorimotor neurons and demyelinating in nature sometimes leading to radiculopathy. Pain and hyperesthesia occur to start with and may be the dominant parts of the neuropathy [113]. Motor neuropathy usually occurs later than sensory neuropathy and can affect the distal extremities first (e.g., foot drop). The best way to target peripheral neuropathy is by targeting the clone by anti-plasma cell therapy. Neuropathy is slow to recover and can take months to a few years after successful eradication of the plasma cell clone. Physical and occupational therapy might have a role in improving the symptoms with stretching, strengthening and balance exercises. Ankle braces, walkers, and wheelchairs should be used as needed in patients. To alleviate the pain of neuropathy, different drugs can be used: gabapentin, pregabalin, amitriptyline, duloxetine, topical lidocaine patches, topical ketamine, lidocaine, etc. [44,114,115]. Neurorehabilitation is an essential part of the treatment plan, and it is essential for sensory as well as motor impairment of the patient. Orthopedic shoes are needed to prevent foot drop. Ankle-foot orthotics (AFO) and other supporting devices are essential in patients with debilitating neuropathy to improve the quality of life of patients. The mainstay of treatment is anti-plasma cell therapy such as IMiD or ASCT which will relieve polyneuropathy in most if not all of patients over times [116].

b.Targeting extravascular fluid overload

The majority of patients will present with signs of fluid overload and anasarca [117], mainly attributed to the activity of serum VEGF and its contribution to capillary leak. Other signs are peripheral edema, ascites, pleural and pericardial effusions [27,118]. In extreme cases third spacing is resistant to diuretics, and patients even may go into renal failure. The serial use of paracentesis and albumin-forced diuresis may in some cases provide benefit for patients [46]. Using anti-VEGF bevacizumab to alleviate third spacing is controversial and should not be used in treating volume overload. When targeting the clone using cyclophosphamide-dexamethasone or bevacizumab, third spacing was not alleviated [19]. In other cases, the combination of treatment using cyclophosphamide, bortezomib, and dexamethasone (CyborD or VCD) alleviated the symptoms dramatically where the patient was back to normal weight, independent of diuretics [83,119].

c.Renal disease

Creatinine values are normal with an approximate of 6% of patients having a value that exceeds 1.5 mg/dL [4] and fewer than 10% of patients have proteinuria of 0.5 g/24 h. Yet, cystatin C, a surrogate renal marker is high in 71% of the patients [95] It is worth noting that some ethnic differences come into play as a study from China addressed the same issue and found that approximately 37% of patients had a creatinine clearance of less than 60 mL/min, 9% less than 30 mL/min, and 15% with microscopic hematuria [42]. The most likely association with renal disease morbidity is the concurrence of Castleman syndrome. Renal disease is highly associated with uncontrollable ascites and anasarca. Sometimes, a significant renal pathology warrants a biopsy of the kidneys with the most common findings being membranoproliferative features and evidence of endothelial injury [58]. Therapy to renal disease specifically is not established, but targeting the clone is the mainstay of treatment with an overall alleviation of signs and symptoms. Renal replacement therapy is seldom necessary.

d.Abnormal pulmonary and heart functions

Pulmonary function tests are abnormal in most of cases, yet respiratory complaints are limited [62,79]. The most common manifestation is pleural effusion; others may be pulmonary hypertension [17], restrictive lung disease, impaired neuromuscular respiratory function, and impaired diffusion capacity of carbon monoxide [28,29,120]. Screening for sleep apnea is indicated in patients who have a severe neuromuscular weakness so that treatment with continuous positive airways pressure or bilevel positive airway pressure can be initiated if necessary. According to He et al., patients with POEMS had subclinical impairment of systolic and diastolic functions of both the right and left heart [79,97,121] as well as left ventricular dysfunction [107,122]. Pulmonary function tests significantly improved in patients’ post-ASCT [33].

e.Organomegaly

Hepatosplenomegaly/lymphadenopathy does not have a mass effect and does not cause local discomfort or require specific therapy. Often, a biopsy of an organ or lymph node is taken during the diagnosis of the disease. The organomegaly and lymphadenopathy resolve with the specific treatment of underlying clonal plasma cell disorder. In up to 30% of the cases Castleman disease can coexist with POEMS syndrome [44]. The approach to organomegaly and general aspect of the disease directed therapy is not different, but addition of IL-6 levels should be measured at diagnosis and followed up during therapy. In this case, elevated IL-6 can be targeted using anti-IL-6 monoclonal antibodies (such as siltuximab), and rituximab can be used for lymphodepletion in addition to the usual anti-plasma cell therapy [122].

f.Endocrinopathy

Endocrinopathy is central in the disease process with hypogonadism being the most common abnormality followed by different endocrinopathies such as thyroid dysfunction, glucose metabolism abnormalities, adrenal insufficiency, and hyperprolactinemia. Above 80% of patients are affected by some form of endocrine dysfunction. For that reason, endocrinopathy is one of the minor criteria in the diagnosis of the disease. In patients with endocrine dysfunction, the appropriate hormone replacement is necessary until anti-plasma cell therapy kicks in. Usually, signs and symptoms decrease after proper chemotherapy regimens. After chemotherapy admission, tapering of thyroid, androgen, and corticosteroid replacement therapy should be considered particularly if hematologic and VEG-F response is achieved [27,123,124,125].

g.Papilledema

About 30% of the patients may present with papilledema, and it may be one of the signs of increased intracranial pressure. Juxtapupillary hemorrhage and optic disc edema may also be seen [126]. Despite patients being asymptomatic for most of the times, a formal ophthalmologic exam is recommended [127], and treatment is necessary when intraocular pressures are very high. The use of acetazolamide and corticosteroids has been effective [128]. In rare cases, where there is an increase in intracranial pressure, serial lumbar punctures may be beneficial. In one case report, ventriculoperitoneal shunt was suggested to be helpful in symptomatic management of hydrocephalous associated with POEMS [129]. Ocular symptoms can also be improved with local treatment using intravitreal bevacizumab and posterior sub-Tenon injection of triamcinolone, with the systemic treatment being the best option for long-term control of the symptoms [130,131,132]. Thalidomide was reported to reduce choroidal thickness and optic disc edema as well [133].

h.Sclerotic bone lesions

Osteosclerotic lesions are found in the majority of cases and are usually confused with bone islands, aneurysmal bone cysts, non-ossifying fibromas, and fibrous dysplasia [134]. The morphology of the lesions differs on CT scans yet FDG-PET)/CT [135] and bone scintigraphy [136] are useful tools in screening for POEMS. If baseline FDG-avidity is present in the PET scan, repeating the scan is useful in monitoring the response to therapy [137]. These lesions typically are non-painful (unless there is a lytic component) and do not usually threaten the integrity of the skeleton. Radiotherapy is one of the mainstays of treatments for <3 bone lesions in POEMS. Additionally, using radiotherapy to lesions that are still FDG-PET-avid after 12 months post-ASCT may be appropriate [46].

i.Targeting skin changes

As previously mentioned, a whole skin exam is necessary since POEMS is recognized as a potential cause of calciphylaxis [138]. Skin changes observed in POEMS do not require a specific treatment, and they all improve after systemic therapy. Calciphylaxis can be devastating, and it has been reported many times in the literature with very poor prognosis [139,140,141]. Bisphosphonates such as etidronate have been shown to be effective in case reports. Local wound care and acetic acid gauze changes and topical potassium permanganate have all been shown to be effective [46].

j.Thrombocytosis, polycythemia, and thrombosis

Thrombosis is prevalent in patients with POEMS reaching as high as 50% of the cases, and it can be the first manifestation of the disease [142]. The hyperplasia of megakaryocytes and even clustering of megakaryocytes can be seen in POEMS syndrome but the myeloproliferation in POEMS is JAK2 negative speaking against a clonal myeloproliferation [34]. Until recently, there were no data that indicated the necessity to lower the platelet count using hydroxyurea or JAK inhibitor ruxolitinib to lower the risk of cardiovascular events [143,144]. In our clinical practice, until the initiation of clone targeting therapy, we recommended to start hydroxyurea (HU) in order to decrease platelet counts, especially in cases where the patient present with thrombotic events [145]. Cerebral and retinal artery infarcts that have recently been reported may be the result of a vasculopathy induced by serum VEGF [146]. It should be noted that a stroke may precede neuropathy in young patients presenting with POEMS [147].

## 7. Response Assessment

The main goal of treatment is to inactivate the plasma cell responsible for this syndrome with the additional goals of preserving and potentially reversing organ dysfunction. The hematologic response and the VEGF response leads to organ responses and symptomatic improvement. For hematologic response assessment International Myeloma Working Group Criteria are used (Table 2) [148].

One of the ways to monitor the hematologic response is to monitor the serum paraprotein at 3–6 months intervals during therapy. Usually, a response is observed 6 months post therapy initiation. Clinicians can assess the patient status using several methods: FDG-PET scan to assess bone disease and organomegaly, and serum VEGF levels (Table 2). To note endocrinopathies should only be assessed periodically if the patient had an abnormal baseline function. VEGF values can be used to assess the response longitudinally in time (Table 2). A clinical response assessment should always be taken separately, despite the hematologic response. It should always incorporate information regarding peripheral neuropathy, organomegaly and fluid status (ascites, edema, effusions). Some use the Overall Neuropathy Limitation Scale (ONLS) score to assess neurologic response [149]. Patients can be categorized according to the following: clinical improvement (IC), clinical progression (PC), mixed clinical response (MC), and clinical stability (SC).

## 8. Prognosis of the Disease

POEMS syndrome is a disease chronic in nature, in which the polyneuropathy can be progressive in nature and correlates with the activity of the plasma cell clone. In comparison to MM patients, POEMS patients have a survival three times longer. A retrospective study revealed that the median survival of the patients with POEMS was 165 months, regardless of how many features of POEMS syndrome are present at the time of diagnosis. However, the patients who, after treatment, still had nail clubbing had a median survival of only 31 months and the patients with extravascular fluid overload had a median survival of 79 months with the two variables being independent of each other. When it comes to the treatments, response to radiotherapy is seen as a good prognostic indicator of survival [42]. Furthermore, concerning mortality, the cause was known to be due to cardiorespiratory failure or recurrent infections and not due to the classic death related to bone marrow failure, hypercalcemia, or pathologic fractures as seen in patients with myeloma [4]. Some patients that died from renal failure had coexistent ascites as well as capillary leak-like syndrome.

A registry for adults with plasma cells dyscrasia (NCT03717844) is set to be completed in 2029. This study will be also looking into patients with POEMS syndrome, and it will assess their baseline status along with their development over time. Patients’ functional statuses, co-morbid medical conditions, cognition, psychological statuses, social functioning, support, medication review, and nutritional statuses will all be reviewed. For any rare disease including POEMS syndrome, registry study can be a very valuable tool for research on the natural history of the disease and the effects of the treatments on the outcomes.

## 9. Conclusions

POEMs syndrome is a multisystem disease with a complex array of signs and symptoms, making the diagnosis, treatment, and follow up of patients challenging. The morbidity and mortality can be reduced with an early diagnosis and prompt treatment. Indications for poor outcomes are fingernail clubbing, respiratory symptoms, and extravascular fluid overload (ascites, pleural effusion, peripheral edema) and lack of complete hematologic response after ASCT. The best therapy has not yet been elucidated through clinical trials, which are challenging to conduct due to the rare nature of the disease. The mainstay of treatment is radiotherapy when there is no bone marrow involvement, and when the disease is limited to fewer than three osteoclerotic bone lesions. When bone marrow involvement is observed, and if the patient is eligible for ASCT, upfront ASCT should be considered. If ASCT is not desired by the patient or they are not eligible due to frailty or multiple organ dysfunction, lenalidomie and dexamethasone is a preferred and well-tolerated regimen. Effective alternatives include proteasome inhibitor boretzomib (though its use may be limited due to baseline neuropathy from the POEMS syndrome). Anti-CD38 immunotherapy daratumumab has shown promising results at the time of relapse in case reports and should be considered in resistant or relapse cases. Further data on its experience are needed for the treatment of POEMS syndrome. Targeting the features of the disease with symptomatic and supportive care as well as emotional support are part of the treatment plan. Follow up and measurement of response are not always easy, because there is no single measure reliable enough. VEGF response usually correlates with disease activity, but M-spike gauzes the hematologic response and provides information on the activity of the underlying plasma cell clone. PET scans can provide valuable information on the activity of the osteosclerotic bone lesions. Adjuvant radiation may be considered if FDG avidity is residual after treatment with ASCT or chemotherapy. If the treatment of the primary lesion was radiotherapy, and there is residual FDG avidity, with a normal VEGF and clinical improvement of the patient, systemic chemotherapy should not be added. The follow up of patients should be completed at least every 3 months for at least several years. It should constitute assessing serial disease symptoms such as peripheral neuropathy, volume status, eyes, skin, and organomegaly as well as of lab values such as M-protein, VEGF, and affected endocrine parameters. Annual pulmonary function tests and bone assessments should be performed as well. After the primary follow up period, all patients should be followed up at least once or twice a year indefinitely. Once the underlying pathogenesis of the disease is better understood, more targeted therapy will be possible.

## Figures and Tables

**Figure 1 jcm-11-07011-f001:**
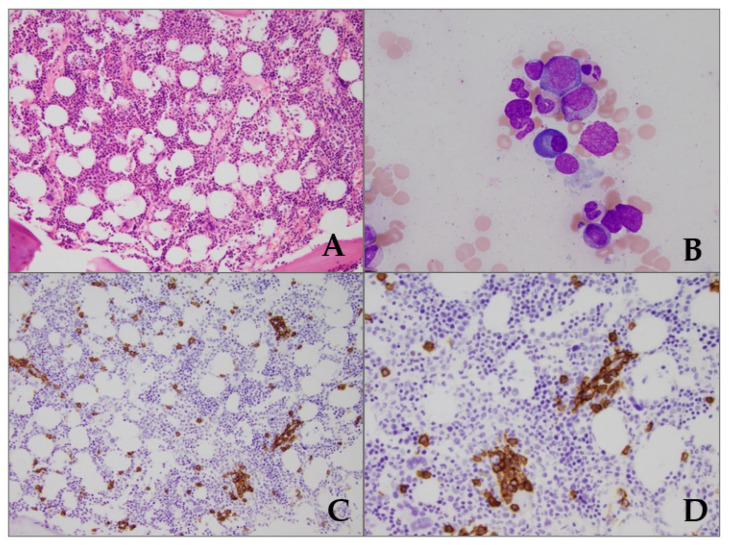
Bone marrow morphologic features in POEMS syndrome of a 64-year-old-man presenting with peripheral polyneuropathy and IgG lambda monoclonal gammopathy. (**A**) Normocellular bone marrow core biopsy for age (Hematoxylin-eosin stain, original magnification ×20). No lymphoid aggregates were observed. (**B**) Bone marrow aspirate smear highlighting a slightly enlarged plasma cell (center) in a background of mixed hematopoiesis. (Wright Giemsa stain, original magnification ×100). (**C**) Plasma cells are slightly increased on the core biopsy estimated at 5% (CD138 immunohistochemical stain, original magnification ×20), which show a perivascular distribution with scattered small clusters. (**D**) Original magnification ×40.

**Figure 2 jcm-11-07011-f002:**
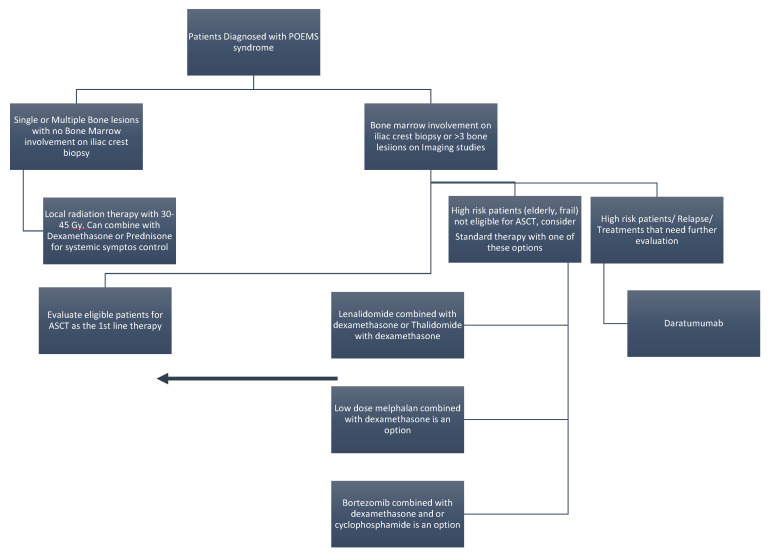
Treatment strategies used in different stages of the disease.

**Table 1 jcm-11-07011-t001:** Mandatory, major, and minor criteria for diagnosis of POEMS.

Diagnostic Criteria for POEMS Syndrome
**Mandatory criteria (Both criteria must be satisfied)**
Polyneuropathy (demyelinating, peripheral, ascending, symmetrical, and sensorimotor) [24]
Monoclonal plasma cell proliferative disorder (almost always λ ^a^)
** *Major criteria (At least one major criterion should be satisfied)* **
Sclerotic bone lesions (densely sclerotic, lytic, or mixed soap bubble) [25,26]
Castleman disease ^b^
An elevated value of serum VEGF ^c^
**Minor criteria (At least one minor criterion should be satisfied)**
Organomegaly (Splenomegaly, hepatomegaly, lymphadenopathy)
Extravascular fluid overload (edema, pleural effusions, ascites) ^f^
Endocrinopathies (Adrenal, thyroid ^d^, pituitary, gonadal, parathyroid, pancreatic) [27]
Skin changes (hyperpigmentation, hemangioma, hypertrichosis, dependent rubor, acrocyanosis, white nails, scleroderma changes, facial atrophy, flushing)
Papilledema (30%), diplopia, and ocular pain [28]
Thrombocytosis/polycythemia ^e^
**Other Symptoms**
Clubbing ^f^, weight loss, hyperhidrosis, pulmonary hypertension, restrictive lung disease, thrombotic diathesis, diarrhea, low vit B12 levels [29]

^a^ Atypical presentation can be IgG kappa. ^b^ Castleman disease variant of POEMS syndrome that occurs without evidence of a clonal plasma cell disorder should be considered separately. ^c^ VEGF level is typically >200 pg/mL in POEMS [30]. ^d^ A high prevalence of diabetes mellitus and thyroid abnormalities is seen in patients not associated with POEMS disease; hence these are considered minor criteria. ^e^ Fifty percent of patients will have bone marrow changes that may help distinguish it from a typical MGUS or MM. ^f^ Anemia and/or thrombocytopenia are atypical in POEMS unless Castleman disease is present. Patients with clubbing had a median survival of 2.6 years those with extravascular fluid overload a had a median survival of 6.6 years.

**Table 2 jcm-11-07011-t002:** Criteria for response assessment.

Hematologic response	CR: Negative bone marrow/Negative immunofixation of serum and urine
	VGPR: 90% reduction in M protein or immunofixation positive only if M protein was at least 0.5 g/dL at baseline
	PR: 50% reduction in M protein or immunofixation positive if baseline M protein was at least 1.0 g/dL
	NR: Less than a PR
VEGF response	CR: Normalization of plasma VEGF <87 pg/mL
	PR: Decrease of ≥50 percent (baseline must be ≥200 pg/mL)
	NR: Less than a PR
Radiologic response by FDG PET	CR: Initial FDG avidity on a baseline PET scan that disappears
	PR: Initial FDG avidity that was 50 percent improved
	NR: Not meeting CR or PR

CR: Complete response; VGPR: Very good partial response; PR: Partial response; NR: No response.

## Data Availability

Not applicable.

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
