# Peer review of "Recent Advances in the Treatment and Supportive Care of POEMS Syndrome"

_jcm, 2022, doi:10.3390/jcm11237011_

Round 1

Reviewer 1 Report

The authors comprehensively have reviewed diagnosis and treatment aspects of POEMS syndrome. Followings are my comments to the authors.

You should specify that concentration of VEGF is largely different between serum and plasma. When discussing VEGF levels, you should clearly state whether you are discussing plasma or serum.

Table 1; You have modified original diagnostic criteria (Dispenzieri., Am J Hematol 2017. Ref#22) to create Table 1. You need to mention the rationale of this modification otherwise you should quate original version.

Algorithm 1; There is no consensus on first line therapy in non-eligible ASCT cases (The authors recommended LDex as a first line). I think thalidomide with dexamethasone can be an option. Isatuximab and CAR-T should not be incorporated into treatment algorithm because of lack of evidence in POEMS.

Table 2, VEGF response; These values indicate plasma VEGF. Please specify.

P6, line 163-165; “The differential diagnoses to keep in mind are chronic inflammatory polyradiculoneuropathy (CIPD), monoclonal gammopathy of undetermined significance (MGUS), and immunoglobulin light chain (AL) amyloid neuropathy.” Chronic inflammatory demyelinating polyradiculoneuropathy (CIDP) is correct. Anti-myelin-associated-glycoprotein (MAG) neuropathy is also one of the important differential diagnoses because it has paraprotein with sensory-motor demyelinating polyneuropathy.

P8, line 234-224; “Induction therapy is not required because of the small clone size of clonal plasma cells and hence a small tumor burden.” In my opinion, the need for induction therapy is inconclusive. Induction therapy has some advantages. First, induction therapy may reduce the incidence of complication (Li et al., Leukemia 2017). Second, normalization of serum VEGF level at ASCT may prolong OS and PFS (Ohwada et al., Blood 2018. Ref#67).

P14, line 441; “For that reason, endocrinopathy is one of the major criteria in the diagnosis of the disease.” This sentence could be confusing to readers because endocrinopathy is one of the minor criteria.

Author Response

Please find attached our replies to the reviewers' comments.

Reviewer 2 Report

Zerdan et al. comprehensively summarize the biology and treatments of POEMS syndrome. The whole manuscript is very well described and it helps readers to understand the current progress of this rare disease. Some minor points could be updated to improve the quality of this review. See comments.

1. In the section of “Pathogenesis of the disease”, the authors show a paper that suggests the possibility that VEGF producing cells are non-clonal plasma cells. However, a recent paper also suggests that VEGF mRNA levels are not elevated in both clonal and non-clonal plasma cells using single cell RNA-seq. Therefore, it still remains unknown which cells are VEGF producing cells. The authors could add some comments about this point.

2. He et al. reported 13 cases of atypical POEMS syndrome without detectable monoclonal gammopathy in Ann Hematol 2019. Although those cases don’t match the current diagnostic criteria which requires the presence of M-proteins, the paper may be worth to be mentioned in the Clinical Presentation and Diagnosis section because the current criteria may not be perfect to diagnose this rare disease.

3. POEMS syndrome shows a high incidence of engraftment syndrome during ASCT and pre-ASCT treatment reduces the incidence of the syndrome among the patients who decrease the level of VEGF. The information is clinically important and needs to be included in the ASCT section.

4. I agree that lenalidomide replaces thalidomide due to the lower incidence of severe toxicities, however, the authors should describe the results of phase 2/3 thalidomide trial, for example, primary endpoint, duration of the response, incidence of severe toxicities etc. Otherwise, it’s hard to say that lenalidomide is superior than thalidomide because no study directly compared those 2 regimens.

Author Response

Please find attached the reviewers' comments.
